# Self-Perceived Clinical Competence of Nurses in Different Working Experiences: A Cross-Sectional Study

**DOI:** 10.3390/healthcare11212808

**Published:** 2023-10-24

**Authors:** Ippolito Notarnicola, Dhurata Ivziku, Daniela Tartaglini, Lucia Filomeno, Raffaella Gualandi, Simona Ricci, Marzia Lommi, Barbara Porcelli, Barbara Raffaele, Graziella Montini, Federica Maria Pia Ferramosca, Erica Di Maria, Anna De Benedictis, Ebru Baysal, Roberto Latina, Gennaro Rocco, Alessandro Stievano

**Affiliations:** 1Centre of Excellence for Nursing Scholarship, OPI of Rome, 00136 Rome, Italy; genna.rocco@gmail.com (G.R.); alessandro.stievano@gmail.com (A.S.); 2Department of Nursing, Catholic University “Our Lady of Good Counsel”, 1000 Tirana, Albania; 3Department of Health Profession, Foundation Policlinic Universitario Campus Bio-Medico, 00128 Rome, Italy; d.ivziku@policlinicocampus.it (D.I.); d.tartaglini@policlinicocampus.it (D.T.); r.gualandi@policlinicocampus.it (R.G.); a.debenedictis@policlinicocampus.it (A.D.B.); 4Department of Biomedicine and Prevetion, University Tor Vergata, 00133 Rome, Italy; lucia.filomeno@uniroma1.it (L.F.); federica_ferramosca@hotmail.it (F.M.P.F.); 5UOC Care to the Person, Local Health Authority Roma 2, 00159 Rome, Italy; siricci2017@gmail.com (S.R.); marzia.lommi@gmail.com (M.L.); barbara.porcelli@aslroma2.it (B.P.); barbara.raffaele@aslroma2.it (B.R.); graziella.montini@aslroma2.it (G.M.); erica.dimaria@aslroma2.it (E.D.M.); 6Department of Fundamentals Nursing, Faculty of Health Sciences, Manisa Celal Bayar University, 45030 Manisa, Turkey; e_bay100@hotmail.com; 7Department of Health Promotion, Mother and Child Care, Internal Medicine and Medical Specialties, University of Palermo, 90128 Palermo, Italy; roberto.latina@unipa.it; 8Department of Clinical and Experimental Medicine, University of Messina, 98100 Messina, Italy

**Keywords:** assessment, competence, nurse competence scale, nurses’ roles

## Abstract

Background: Competence is an essential concept for measuring nurses’ performance in terms of effectiveness and quality. To this end, our analysis highlighted the process of acquiring competencies, their self-evaluation into clinical practice, and how their proficiency levels change throughout the nursing career. In detail, this research explored nurses’ perceived level of competence and the factors that influence it in different contexts. Methods: A cross-sectional survey using a structured questionnaire to assess the nursing participants’ perception of their competencies in different clinical settings was accomplished. Results: A descriptive and bivariate analysis was performed on 431 nurses. Most respondents assessed their level of competence to be higher than their roles required. The Kruskal–Wallis test confirmed that nursing experience was a relevant factor influencing nursing competencies. Conclusions: We suggest improving the competence of practicing nurses, using experience as a measurable effect of their development.

## 1. Introduction

Nursing competence is a mandatory issue for national and international stakeholders alike. For example, Italy follows directives issued by the World Health Organization, the International Council of Nurses, the Council of the European Union, and Italy’s Ministry of Education, which define the objectives for developing nursing professionals. These objectives address the required competencies for quality nursing care, educational qualifications, obligatory standards of practice for professional nurses, and ongoing competence development [1,2].

Competencies are fundamental to nursing because they can guarantee quality, safety, and health assistance and save on health costs; therefore, establishing their reliable measurement is crucial for their effectiveness [3]. In other words, identifying fundamental, practical competencies reflects the reality of patient care and professional practice [4].

### Background

Benner [5] defines competence, in seminal works, as developing a task with desirable results in various real-world circumstances. According to some authors, many components co-build nursing competencies, principally the knowledge acquired from cognitive abilities [6,7]. This knowledge involves mastering the mass of information on which the nursing practice is founded. Nurses need to put factual knowledge into practice in an actual clinical situation [6]. Therefore, assessing nursing competencies with a reliable tool such as the Nurse Competence Scale (NCS) is vital for evaluating a nurse’s abilities [8,9,10]. 

Our survey findings may be fundamental to those countries introducing nursing development and professional accreditation programmes, qualification examinations, and quality control evaluation, as these are considered fundamental by international studies that have implemented the NCS in their context [11]. 

The most significant difficulty in evaluating clinical competencies rests in defining the term competence; in fact, competence is still a vague concept, defined differently by various stakeholders [12]. An analysis of some of these definitions highlighted crucial features of the construct but still did not define it satisfactorily—probably due to its complex, multidimensional, and multidisciplinary nature [13].

For example, clinical nursing skills should consist of certain activities performed in clinical practice, such as providing care, managing complex situations, ensuring the quality of care provided, and nursing research. 

Dellai et al. [14] state that nursing competence assessments are a prerequisite for ensuring that nurses are qualified for patient care and for identifying areas for developing nursing practice. The evaluation of nurses is essential, requiring a continuous review of the necessary competencies to identify patient needs and a systematic activation of self-assessment processes, and the capacity to maintain a high standard of care [15,16,17]. Although the discussion concerns nursing competencies and quality of care, Istomina et al. [18] contend that there is no relationship between them. With respect to levels of competence, Benner believes that these can be described as a range of nursing expertise that goes from novice to expert, where the achievement of competence depends on continuous training in clinical settings [19]. In line with this perspective, exploring the relationship between the level of competence achieved, experience gained, and care outcomes is central. These three elements offer the foundation for obtaining information on improving education and nursing practice.

The concept of competence has also been extensively analysed and discussed by many authors [8,20]. For example, Meretoja et al. [21] have defined such competencies through three perspectives: the ability to practice in a specific role, the integration of knowledge and clinical competencies within the emotional relationships of clinical practice, and, eventually, professional development through clinical experience [21,22]. 

Several tools for evaluating clinical competencies have been developed in the nursing discipline. These include the NCS [23], the Competency Inventory for Registered Nurses [24], the European Questionnaire Tool [25], and the Holistic Nursing Competence Scale [26]. The NCS was utilized in our study [23]. 

In fact, nurses’ competence is usually measured using the NCS [23]. The NCS tool has been used in international research for over a decade [8]. Other countries have since implemented the NCS test, including Australia, England, Switzerland and Italy. In Italy, the NCS received linguistic and cultural validation by Finotto and Cantarelli [27].

Based on earlier research by Meretoja et al. [23], the NCS was developed in Finland in 2007 as a tool for nurses to self-assess their clinical competencies. The NCS was tested in different hospital settings (e.g., emergency, intensive care, operating rooms, medicine, surgery, neurology, and psychiatry) and was utilized with novice and experienced nurses [1,8,15]. 

In the literature related to the assessment of nursing competencies using the NCS, the categories with the highest level of competence included helping role, managing situations, diagnostic function, and work role, whereas the lowest level of competence was found in ensuring quality, therapeutic intervention and teaching/coaching [1]. According to some authors in several studies that have implemented the NCS, certain factors were found to influence a high level of competence, such as age, years of work experience, length of service in the same clinical environment, higher education, previous professional qualification, experience in healthcare facilities, and job satisfaction [8,18,28]. Other studies have evaluated the correlation between the level of competence and other variables, such as the frequency of using these competencies within different clinical settings and nurses’ working experience [18,29,30]. 

Therefore, in this complex framework, this study aims to analyze the perception of nurses’ competencies and their frequency in different healthcare contexts through the NCS, a tool based on the conceptual model proposed by Benner.

## 2. Materials and Methods

A cross-sectional survey using a structured questionnaire to assess the nursing participants’ perception of their competencies in different clinical settings was conducted. The study was performed in several hospitals situated in Italy. Data were collected from April to September 2019. The Strengthening Reporting of Observational Studies in Epidemiology (STROBE) [31] guidelines were used to ensure quality reporting. A cluster sample method was used because of its suitability for large target populations. The sample consisted of nurses who were employed in different clinical settings at public and private healthcare facilities. Eligible respondents were included in the study if they (i) were registered nurses, and (ii) had worked in hospitals, private clinics, and community facilities for at least three months. All respondents who returned the questionnaire having completed at least 70% of the items were included in the study.

The 73-item NCS comprises seven dimensions or categories of competence that were developed based on the nursing practice framework defined by Benner [5]. Each dimension of the NCS consists of different sets of core competencies, distributed as follows: helping role (7 items); teaching/coaching (16 items); diagnostic functions (7 items); managing situations (8 items); therapeutic interventions (10 items); ensuring quality (6 items); and work role (19 items).

The NCS consists of two separate tools for self-assessing competencies: a visual analogue scale (VAS) and a Likert scale (LS), administered at specific points during the study. The VAS tool measures the level of self-perceived nursing competencies within a range of 0–100, where a score of zero (VAS = 0) indicates the lowest level of competence and a score of one hundred (VAS = 100) is the highest. For descriptive purposes, the average values obtained as a result of VAS measurements were stratified into four categories to express a judgment about the level of nursing competencies achieved, designated as follows: low (VAS = 0–25), quite good (VAS > 25–50), good (VAS > 50–75) and very good (VAS > 75–100) [14]. The Likert scale tool was applied to measure the nurses’ self-assessment of how frequently they used their competencies in different clinical settings in their daily professional performance duties. The latter measure was obtained using a 4-point LS (LS 1 = not applicable in my work, LS 2 = very rarely, LS 3 = occasionally, and LS 4 = very often applicable in my work).

This study used the version of the NCS translated into Italian by Finotto and Cantarelli [27]. The internal consistency of the NCS components was analyzed by calculating Cronbach’s alpha and the stability was assessed using the intraclass correlation coefficient (ICC), which were assumed to be equivalent when applied to the same data. These indicators admit values in a range of 0–1, where the closer the value is to 1, the stronger the relationship will be among the LS and VAS average scores related to the set of items included in each dimension of the NCS. Table 1 shows that Cronbach’s alpha and the ICC of the VAS average scores display values between 0.91 and 0.96, which reveals an excellent correlation level for all NCS dimensions. Table 1 also shows Cronbach’s alpha and the LS’s ICC, reporting values between 0.87 and 0.94. These too reveal an excellent correlation level in all competence classes. The highest score was obtained in the teaching/coaching dimension (a = 0.94; ICC = 0.94), while the lowest was recorded in the dimensions of managing situations and diagnostic function (a = 0.86; ICC = 0.86). The Cronbach’s alpha results were between 0.80 and 0.96, confirming the NCS’s reliability, which supports similar findings in other studies [27,28,32].

### 2.1. Study Design and Data Collection Procedures

A total of 800 questionnaires were sent to participating hospitals between April and September 2019. All study participants were permanent nursing staff members; all were directly involved in patient care. Participating hospitals were sent the questionnaire, a consent form, and a self-addressed return envelope. Data were collected using anonymous, self-administered, and structured questionnaires. The completed questionnaires and consent forms were sealed and returned in self-addressed return envelopes. All completed questionnaires were protected confidentially, without identifiable tags or specific personal information.

### 2.2. Sample Size

This study aimed to reach as many participants as possible and collect as much data as feasible [33]. A sample size calculator was used to determine the representative target sample size required to meet the study’s objectives and to have adequate statistical power [34]. The sample size calculator calculated a required size of 377 participants, based on a 50% response distribution, a 4% margin of error, and a 99% confidence level.

### 2.3. Data Analysis

Data analyses were conducted using SPSS 26.0 for Windows and entailed a descriptive and multivariate analysis. The results of the descriptive analysis were expressed as the mean values of the socio-demographic data, while the VAS and LS average scores related to the seven NCS dimensions. Although we used VAS averages and LS, we mainly developed VAS-based analyses. In addition, the latter analysis consisted of observing the VAS mean values when stratified according to years of service and the frequency of using competencies. The correlation between the LS and VAS measures was considered by calculating the non-parametric Spearman’s rho without separating variables. In addition, a normality test was applied to the data set, whereas the Kruskal–Wallis test calculated the associations between variables. 

### 2.4. Ethical Consideration

This study was ethically approved by the Center of Excellence for Nursing Scholarship OPI Rome, protocol number 2.19.08, following international ethical principles and Italian legal and research ethics requirements for non-interventional studies.

The enrolment of eligible nurses was voluntary after they were provided with all the information about the survey, which included expressing their consent to participate. Written informed consent was also obtained from all participants. Nurses who agreed were guaranteed confidentiality during all phases of the research; consequently, all data were anonymized and aggregated. The study was designed, conducted, recorded, and reported on consistent with the international ethical and scientific quality standards indicated by good clinical practice (GCP) and standard operating procedures (SOPs). Participants were asked to provide written informed consent.

## 3. Results

The questionnaire was administered to 701 nurses, and 449 (64.14%) completed it. Eighteen questionnaires (4.01%) were rejected for incomplete data. The response rate was 61.57% (*n* = 431).

Table 2 shows the socio-demographic data. Among the respondents, 92.58% were nurses, 6.03% were nurse coordinators, and only 0.7% were nurse managers. The median age of the participants was 44 years. Considering the years of work experience, 70.3% had been employed for over 10 years, 11.37% declared a length of service of 0–3 years, and 15.78% reported a 4–10-year range. Respondents were mainly employed in the following areas of work: medicine (*n* = 94; 21.81%), intensive care (*n* = 82; 19.03%), surgical wards (*n* = 80; 18.56%), and emergency settings (*n* = 78; 18.1%).

Some 35.5% of the nurses surveyed (*n* = 153) reported having undertaken further education after their fundamental training. Of this group, 27.5% had undertaken a specialization course, 3.94% possessed a master’s degree, 2.78% had another degree in other fields, and 64.5% had no further education.

The average VAS scores that describe the nurses’ perceived level of competence are presented in Table 3, which shows high mean values in all the dimensions of competence (VAS = 72.6–77.9). According to the scale proposed by Dellai et al. [14], these results correspond to the level of competence registered between good and excellent; in fact, the average value obtained from the overall observations was VAS = 74.8, namely, very good. The analysis reported that the highest average score (VAS = 77.9) was found in the dimension managing situations, while the lowest average value (VAS = 72.9) was measured in the dimensions ensuring quality and helping role.

As shown in Table 3, by stratifying the level of perceived nursing competence expressed as VAS scores according to the frequency of utilization of the competence, we found that the highest VAS values were detected in the LS scores of very often (VAS = 79.7–83.3). The latter measures corresponded to a VAS mean value (VAS = 81.8) significantly higher than the VAS average score (VAS = 74.8) determined across all the competence categories.

We could detect that when stratified by the frequency of competencies being put into practice very often, the respondents’ perceived level of competence was highest in the dimension work role (with a VAS score of 83.3), followed by the dimension therapeutic interventions (with a VAS score of 82.4) and the dimension managing situations (with a VAS score of 81.8). Lastly, a slight difference was found between the scores detected in the dimensions diagnostic function, ensuring quality and teaching/coaching, which registered at VAS = 80.4, VAS = 80.1, and VAS = 80.9. The helping role dimension had the lowest score (VAS = 79.7).

In Table 4, we selected the variable linked to the nurses’ years of work experience, and we further stratified the VAS average scores of our sample into the following classes: 0–3 years (VAS = 68.4), 4–10 years (VAS = 76.6), and over ten years of experience (VAS = 75.5). To analyze this aspect, the increase in working years was associated with a direct rise in the LS’s average values. The observable average of the LS scores for each class of competence grew in those who self-evaluated as using the competence very often (0–3 years (VAS = 76.7); 4–10 years (VAS = 81.5); over ten years of experience (VAS = 82.5)), and in those who occasionally used their competencies (0–3 years (VAS = 67.2); 4–10 years (VAS = 78.2); over ten years of experience (VAS = 74.9)). If one looks at the findings related to the frequency of using competencies, the highest average LS value was found in the dimension managing situations, with a score of 3.36, while the lowest was measured in the ensuring quality dimension, with a score of 2.98.

Furthermore, we found a moderately positive Spearman’s correlation with a high statistical significance (*p* < 0.001) between the level of perceived competence and the frequency of its use in clinical practice for the work role dimension (Spearman’s rho = 0.501). Conversely, the managing situations dimension showed a low positive Spearman’s correlation (Spearman’s rho = 0.377).

Finally, in Table 4, a Kruskal–Wallis H test was conducted to determine if the VAS average scores were different for the three groups whose length of working experience was either (a) 0–3 years (*n* = 49), (b) 4–10 years (*n* = 68), or (c) over ten years (*n* = 303). The Kruskal–Wallis H test showed a statistically significant difference in the self-perceived level of competence among the three groups regarding the seven NCS dimensions (*p* < 0.05). A particularly noteworthy finding entailed the significant VAS score differences among the three groups selected, namely, 0–3 years of experience (VAS = 68.4), 4–10 years of experience (VAS = 76.6), and over ten years of experience (VAS = 75.5).

## 4. Discussion

Our findings mainly quantify the results of two variables that display nurses’ perceptions about their competence levels and the frequency with which they use various nursing competencies in different clinical settings. We highlighted the extent of correlation between the two variables and identified factors associated with a high level of competence.

Nurses reported a high level of self-perceived competence across the seven NCS dimensions, seeing themselves as very good (VAS = 74.8). The highest average VAS score, namely, 77.9, was found in the dimension of managing situations. Our results are supported by those of O’Leary [35], who investigated self-reported competence levels in a sample of nurses with 15 years of work experience. O’Leary [35] also found the highest VAS score (81.9) in the managing situations dimension. According to Meretoja et al. [21], these findings mean that nurses can recognize unstable situations and prioritize activities flexibly and appropriately, promoting cooperation and choosing different solutions. Moreover, according to some studies [11,32], nurses at the top of their career achieve the highest scores in the dimension of managing situations and score the lowest in ensuring quality. 

Concerning nurses at the beginning of their work experience (0–3 years), our results are consistent with those of Wangesteen et al. [8], which showed the lowest scores in the NCS dimension of ensuring quality, with a score lower than ours (respectively, VAS = 53.8 and VAS = 65.3). The ensuring quality dimension of competence aims to evaluate results and contribute to patient care development [23]. 

Lima et al. [36] and Wangesteen et al. [8] found that the helping role dimension scored highest in all the competence self-perceptions of beginner nurses (respectively, VAS = 84.4 and VAS = 70.0). On the other hand, our results tended to have the lowest score in this dimension, both in the overall average mean (VAS = 72.9) and for the novice nurses (VAS = 65.6). The helping role dimension consists of those competencies intended to help the patient cope with problems and provide ethical and individualized care. Our results align with Lima et al. [36], who argue that it would be interesting to know how other studies have implemented the NCS regarding developing the helping role dimension.

We analyzed the trends in the level of competence perceived by nurses according to the frequency with which they used their competencies on the one hand and their work experience on the other. Our results reveal that nurses with over ten years of experience who used their competencies more frequently showed a higher overall level of competence than nurses with the same experience who declared they used their competencies occasionally. This confirms that experience—the notion on which Benner founded his theory of the NCS—is fundamental to developing skills [5]. In fact, as described in Table 3, we see that the class of nurses that defined themselves as having a higher level of competence were those with 4–10 years of work experience and not, as could reasonably be expected, those with over ten years of experience. Moreover, the level of competence and years of work experience follow a direct relationship only if we consider the class of nurses who employ their competencies with the highest frequency. In contrast, this direct relationship does not apply to the class of nurses who applied their nursing competence occasionally in various clinical settings. Even in nurses with years of experience in the 0–3 range, the trend in competencies could be variable; for example, nurses who have worked for three months may rank less in self-perceived competence and the frequency of use of competencies.

According to our findings, the degree to which nursing competencies are put into practice corresponds, on average, to a range of 3–4 (LS = 3.16) in all seven NCS dimensions. Moreover, the highest average score was found in the dimension of managing situations (LS = 3.36), which is consistent with the findings by Bahreini et al. [32], whereas the lowest value was detected in the ensuring quality category (LS = 2.98). This last result is also supported by O’Leary’s [35] study, which found a score of 55%.

Furthermore, our analysis aimed to observe trends in competence levels when stratified according to the frequency of use across the various NCS dimensions. Our findings suggest a direct relationship obtained between the two variables, that is, the nurses’ perceptions of their competence increased in value (from VAS = 74.1 to VAS = 81.8) when an intensive use of nursing competencies was observed in clinical settings (respectively, from LS = 2.5–3.4 to LS = 3.5–4.0). These findings align with O’Leary [35], who showed a linear trend between the two variables for almost all the NCS dimensions.

Our following observation related to the nurses’ perception of their competence levels after the levels were stratified according to the frequency of applying their competencies and the length of their nursing experience in years. We found that competence levels increased directly related to working experience when the frequency of competence use was very high (LS = 3.5–4.0). Thus, in line with the study conducted by Numminen et al. [1], the more work experience nurses gain, the more likely the difference between the level of their perceived competence and their effective use of such competence in practice will narrow. Our research confirmed this result by focusing on stratified competence levels according to the highest frequency of competence use. In contrast, several studies reported that the trend between the self-assessed level of competence and working experience tended to consolidate, remaining at the same point after a certain level of expertise had been gained during the nursing career [1,26]. 

We also explored the role of expertise in the development of nursing competence. For this, we used a Kruskal–Wallis test to determine whether the perceived level of competence would change among the groups of nurses with different degrees of expertise across the NCS dimensions. The Kruskal–Wallis test’s *p* values (<0.05) revealed a difference in the VAS average scores among the groups of nurses stratified by level of expertise. Thus, the perceived level of competence was lower for registered nurses at the beginning of their careers and higher for nurses with at least four years of service, whereas it remained at approximately the same level for the more experienced nurses. By comparison, other research shows a positive, albeit not strong, correlation between experience and level of competence; cases in point are Meretoja et al. [21] and Salonen et al. [29], whose results were R = 0.303–0.337 and R = 0.272 (*p* < 0.001), respectively. O’Leary [35] supports the latter results, finding a correlation of R = 0.27 (*p* < 0.05). These findings are consistent with ours and also with several international studies [1,15]. It is relevant to highlight the role that experience, as a measurable effect, plays in developing nurses’ clinical competencies [1,15,35]. 

Lastly, the Spearman correlation was enlisted to determine the degree of association between the perceived level of competence and the frequency of competence use. The results suggested that these two variables were directly related through a positive and linear trend. This signifies that nurses who report high or low values in a variable tend to report, respectively, high or low values in the second variable. The Spearman coefficient resulted in a range of low scores (Spearman’s rho of 0.377–0.430) across the seven NCS dimensions except for the work role category, where a moderately positive association (Spearman’s rho of 0.501) was found. Therefore, because of a low to moderate correlation strength, the dependent variable’s value could not be derived from the value of the independent one, as if it was following a perfect linear model. Nevertheless, a positive correlation between the perceived level of competence and frequency of competence use was found in several international studies, such as Meretoja et al. [21], Salonen et al. [27], and Numminen et al. [1]. Numminen et al. [1] argue that this correlation is crucial for proving nurses’ responsibility for acting in line with their perceived competencies. 

### 4.1. Limitations

In our study, some limitations must be considered when making sense of the results. This study used a self-completed questionnaire and was limited by the accuracy of participants’ responses. Participation was voluntary and, therefore, dependent on people agreeing to participate in the study. Participants were enrolled using convenience sampling, and data were collected through a cross-sectional approach. For this reason, some caution is required when generalizing the results; the generalizability of the results may not apply to nursing students, for example. In addition, reliability was assessed only by assessing the internal consistency of the NCS size, without stability information. Furthermore, our study sample related to only one region in Italy, which needs to be revised to represent the entire Italian nursing context. More than that, attrition bias could have happened when participants did not respond to specific questions.

We did not ask the participants for their year of graduation; this could have influenced self-perceived clinical competence, as they could demonstrate that they hold less experience of working in the same way; this aspect could be better explored in future studies. 

According to Polit and Beck [37], if random-cluster-type sampling had been used, it would have increased our findings’ validity. Thus, although 431 nurses participated in the study, which constituted an adequate sample size [37], a larger sample size would have strengthened the research outcomes. Moreover, having nurses assess their competencies raised the risk of compromising the objectivity of the NCS measurements, i.e., they could result from overestimating the level of competence evaluated [8]. In addition, a mixed-method study could have better explored the nurses’ perceptions of their competencies. For this reason, future research should examine the indices of stability of the NCS over time.

### 4.2. Implications for Nursing Practice

Competence is an essential concept for measuring nurses’ performance, in addition to being a way of providing effective and quality healthcare services. Our analysis highlighted the process of acquiring competencies and their incorporation into clinical practice, as well as how levels of competence change throughout the nursing career. Our survey findings may be relevant to countries that intend to institutionalize the nursing profession and that plan on introducing a health standards control policy to enhance the quality of healthcare services. 

Finally, educators should also consider introducing an assessment of the clinical skills that nursing students acquire during their university careers.

## 5. Conclusions

For health services to be effective and of high quality, nurses must have adequate skills; for this reason, it is essential to measure them and evaluate their performance in the clinical and professional fields with adequate, reliable tools such as the NCS. 

In this study, the NCS’s implementation in a sample of nurses provided information about their professional development. Targeting the evaluation of competencies remains a critical issue for healthcare organizations. To this end, our analysis highlighted the process of acquiring competencies and their incorporation into clinical practice and how levels of competence change throughout the nursing career.

Furthermore, the reported assessments of competence levels among nurses with different degrees of work experience might contribute to the debate concerning possible ways to develop nursing competencies in clinical practice according to the individuals’ expertise. Therefore, our investigation can be helpful for policymakers to change the competence standards of nurses in different clinical settings. Furthermore, it can be helpful to educators so that they can adapt university courses to continuously evolving skills.

## Figures and Tables

**Table 1 healthcare-11-02808-t001:** Cronbach’s alpha (α) and intraclass correlation coefficient (ICC) calculated for VAS and Likert scale average scores of the seven dimensions of the NCS (*n* =431).

	VAS Scale			Likert Scale		
Dimension of the NCS	α *	ICC	95% CI	α *	ICC	95% CI
Helping Role	0.93	0.93	0.92–0.94	0.87	0.87	0.85–0.89
Teaching Coaching	0.96	0.96	0.95–0.96	0.94	0.94	0.93–0.94
Diagnostic Functions	0.91	0.91	0.90–0.93	0.86	0.86	0.84–0.88
Managing Situations	0.92	0.92	0.91–0.93	0.86	0.86	0.84–0.88
Therapeutic Interventions	0.95	0.95	0.94–0.95	0.91	0.91	0.90–0.93
Ensuring Quality	0.94	0.95	0.94–0.95	0.89	0.85	0.87–0.91
Work Role	0.96	0.96	0.95–0.96	0.92	0.92	0.91–0.93

* Coefficient based on the standardization of the items.

**Table 2 healthcare-11-02808-t002:** Participants’ socio-demographic data (*n* = 431).

Characteristic	N	%
Gender		
Male	158	36.66
Female	270	62.65
Undeclared	3	0.7
Age (years)		
Average (SD)	43.8 (10.6)	
Title		
Nurse	399	92.58
Nurse Coordinator	26	6.03
Nurse Manager	3	0.7
Undeclared	3	0.7
Work Settings Investigated		
Medicine	94	21.81
Critical Area	82	19.03
Surgical Area	80	18.56
Emergency/Urgency Area	78	18.1
Ambulatory	31	7.19
Other	29	6.73
Undeclared	37	8.58
Basic Education		
Professional Degree	217	50.35
University Degree Courses	207	47.49
Undeclared	7	1.62
Post-Basic Education		
None	7	0.46
Specialist Postgraduate Course	117	27.5
Master’s Degree	17	3.94
Master’s Degree in Other Fields	12	2.78
Undeclared	278	64.5
Years of Service		
Average (SD)	19.08 (11.02)	
0–3 years	49	11.37
4–10 years	68	15.78
>10 years	303	70.30
Undeclared	11	2.55
Years of Service in the Same Operative Unit		
Average (SD)	9.2 (7.7)	

**Table 3 healthcare-11-02808-t003:** VAS averages stratified based on the frequency with which they were used (*n* = 431).

Dimension of the NCS	Average *	SD	Very Often (LS Response from 3.5 to 4)	Occasionally (LS Response from 2.5 to 3.4)
			Average (SD)	Average (SD)
Overall	74.8	13.8	81.8 (9.2)	74.1 (11.4)
Helping Role	72.9	15.7	79.7 (10.4)	70.7 (14.9)
Teaching Coaching	73.7	15.0	80.9 (9.5)	73.5 (11.7)
Diagnostic Functions	72.6	16.2	80.4 (10.7)	71.9 (13.5)
Managing Situations	77.9	13.7	81.8 (10.4)	75.3 (14.3)
Therapeutic Interventions	76.1	15.0	82.4 (10.2)	73.6 (13.7)
Ensuring Quality	72.9	17.4	80.1 (11.7)	73.5 (12.7)
Work Role	77.6	13.5	83.3 (10.2)	75.3 (12.1)

* VAS = 0–25, low; 25–50, quite good; 51–75, good; 76–100, very good.

**Table 4 healthcare-11-02808-t004:** VAS averages stratified based on frequency of use of competencies and years of service (*n* = 431).

Dimension of the NCS	Average	SD	Very Often	Occasionally	*p*-Value *
			(LS Response from 3.5 to 4)	(LS Response from 2.5 to 3.4)	
			Average (SD)	Average (SD)	
Years of service 0–3; *n* = 49					
Overall	68.4	11.6	76.7 (7.7)	67.2 (11.3)	<0.001
Helping Role	65.6	13.8	70.0 (8.1)	64.9 (13.5)	<0.001
Teaching Coaching	69.2	12.6	76.6 (7.7)	67.9 (11.0)	0.003
Diagnostic Functions	67.1	12.8	71.8 (12.8)	65.2 (12.1)	0.001
Managing Situations	71.2	12.5	77.1 (10.5)	66.7 (11.7)	<0.001
Therapeutic Interventions	68.7	13.0	80.0 (9.1)	64.6 (9.7)	<0.001
Ensuring Quality	65.3	15.3	78.3 (12.1)	65.1 (11.8)	<0.001
Work Role	71.8	11.7	76.1 (11.9)	70.7 (11.3)	<0.001
Years of service 4–10; *n* = 68					
Overall	76.6	15.0	81.5 (10.1)	78.2 (9.0)	<0.001
Helping Role	75.3	15.1	79.4 (12.0)	76.6 (11.5)	<0.001
Teaching Coaching	75.4	16.2	79.8 (9.5)	76.5 (11.6)	0.003
Diagnostic Functions	74.9	16.2	80.4 (11.3)	75.4 (10.3)	0.001
Managing Situations	79.7	13.6	83.5 (10.3)	78.0 (14.0)	<0.001
Therapeutic Interventions	77.9	16.3	84.7 (10.2)	76.4 (10.4)	<0.001
Ensuring Quality	73.6	19.9	82.4 (11.3)	71.8 (12.9)	<0.001
Work Role	79.0	15.3	84.5 (9.7)	78.3 (11.0)	<0.001
Years of Service >10; *n* = 303					
Overall	75.5	13.8	82.5 (9.0)	74.9 (11.4)	<0.001
Helping Role	73.6	15.8	80.9 (9.7)	70.7 (15.2)	<0.001
Teaching Coaching	73.9	15.1	81.7 (9.6)	74.2 (11.5)	0.003
Diagnostic Functions	73.0	16.7	81.6 (10.1)	72.5 (14.2)	0.001
Managing Situations	78.8	13.7	82.3 (10.3)	76.4 (14.5)	<0.001
Therapeutic Interventions	76.9	14.9	82.2 (10.4)	74.7 (14.7)	<0.001
Ensuring Quality	73.9	17.1	79.9 (12.0)	76.0 (12.1)	<0.001
Work Role	78.3	13.3	83.9 (10.0)	75.7 (12.3)	<0.001

* Kruskal–Wallis test.

## Data Availability

Not applicable.

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
