# Peer review of "Self-Perceived Clinical Competence of Nurses in Different Working Experiences: A Cross-Sectional Study"

_healthcare, 2023, doi:10.3390/healthcare11212808_

Round 1

Reviewer 1 Report

Thank you for the opportunity to review this paper. This study investigated Self-perceived clinical competence of nurses in different clinical settings. The introduction and methods are well described; however, some concerns should be clearly discussed.

1.The research title seemed not go along with the discussion, the author discussed that nurse’s competence was correlated with frequency of using it (as imply to nursing experience) So, the title may be adjusted toSelf-perceived clinical competence of nurses in different working  experience: A cross-sectional study”.

2.Year of graduation may also affect self-perceived clinical competence. (Meaning: has less experience of working in the same way) If the author did not collect this data, then should mention in the limitation.

3.In the table 2, 3, the author should report not only the average level but also the percent in each level as well (: low n(%), quite good n(%), good n(%), very good n(%))

4.In the table 4, a range of 0-3 year of service may have variation in this group due to inclusion of working experience for at least 3 months. For example, the 3 month-working nurses may rank less in both self-perceived competency and frequency of competency use. The author should mention this in the discussion.

5. Given the conclusions of the study, the authors should provide prospective comments about how this study applies to the practice or nurse training. For example, nurse competency should be assurance by using EPA (entrustable professional activities) evaluation during the nurse’s training, etc.

Author Response

Reviewer 1

Thank you for the opportunity to review this paper. This study investigated the Self-perceived clinical competence of nurses in different clinical settings. The introduction and methods are well described; however, some concerns should be clearly discussed.

1.The research title seemed not to go along with the discussion; the author discussed that a nurse's competence was correlated with the frequency of using it (as implied to nursing experience); the title may be adjusted to "Self-perceived clinical competence of nurses in different working experience: A cross-sectional study".

  • Thank you for the recommendation. We changed the title as suggested.

2.Year of graduation may also affect self-perceived clinical competence. (Meaning: has less experience of working in the same way) If the author did not collect this data, then should mention in the limitation.

  • As recommended by the reviewer, we have added a paragraph in the limitations on this issue.

3.In the table 2, 3, the author should report not only the average level but also the percent in each level as well (: low n(%), quite good n(%), good n(%), very good n(%))

  • While Table 2 treats just socio-demographic data and it is challenging to add ratios, as recommended, in Table 3, we added some considerations on the ratios in each level.

4.In the table 4, a range of 0-3 year of service may have variation in this group due to inclusion of working experience for at least 3 months. For example, the 3 month-working nurses may rank less in both self-perceived competency and frequency of competency use. The author should mention this in the discussion.

  • We acknowledge the suggestion; we have inserted a paragraph in the discussion that clarifies this point to better understand such variations.
  1. Given the conclusions of the study, the authors should provide prospective comments about how thisstudy applies to the practice or nurse training. For example, nurse competency should be assessed by using EPA (entrustable professional activities) evaluation during the nurse's training, etc.
  • We thank the reviewer for the suggestion; we have inserted a paragraph on this theme in the section "future nursing implications" also clarifying the evaluation of university students.

Reviewer 2 Report

Competence in nursing is a crucial concept hence this article is important, and you did a good job undertaking this study. Your analysis was detailed. 

In your abstract under "Background", lines  28-30, you indicated that your analysis highlighted 'integration of (competence) into practice'. I do not recall seeing this part in your analysis and discussion.

I like to draw your attention to  Lines 125 to 126 related to citing authors. Need to be consistent with the rest of the paper.  Likewise on Line 95, you need to recheck  "Based on their earlier research, [23] developed..." [23] cannot replace the names of authors. 

Do you have any theoretical framework for your study? It will be good to identify one as it gives your work more structure. 

Overall, there is need for major editing by an English Language editor to provide the readers with more clarity.  Meanwhile, do note the following:

Line 59-NCS, what does NCS mean? Write the full meaning followed by abbreviation, at the first mentioning. In line with this, do revisit line 112 to correct the error.

Sentence on lines 99 to100 is unclear.

Author Response

Reviewer 2

Competence in nursing is a crucial concept hence this article is important, and you did a good job undertaking this study. Your analysis was detailed. 

In your abstract under "Background", lines  28-30, you indicated that your analysis highlighted 'integration of (competence) into practice'. I do not recall seeing this part in your analysis and discussion.

  • We thank the reviewer for this advice; we apologize; there was a typo that was corrected.

I like to draw your attention to  Lines 125 to 126 related to citing authors. Need to be consistent with the rest of the paper.  Likewise on Line 95, you need to recheck  "Based on their earlier research, [23] developed..." [23] cannot replace the names of authors. 

  • We thank the reviewer for suggesting we put the authors' names to improve the understanding of the paragraph.

Do you have any theoretical framework for your study? It will be good to identify one as it gives your work more structure.

  • Our frame of reference is that of Benner, the same one used in the main NCS study by Meretoja et al. 2003 and 2004.

Overall, there is need for major editing by an English Language editor to provide the readers with more clarity.  Meanwhile, do note the following:

Line 59-NCS, what does NCS mean? Write the full meaning followed by abbreviation, at the first mentioning. In line with this, do revisit line 112 to correct the error.

  • We thank the reviewer for the suggestion we specified the acronym as suggested by the reviewer. Besides, an official translator checked the paper for editing typos.

The sentence on lines 99 to100 is unclear.

  • We thank the reviewer for suggesting. We inserted the author to better clarify the paragraph.

Reviewer 3 Report

The manuscript is about the Self-perceived clinical competence of nurses in different clini-2 cal settings: A cross-sectional study.

The manuscript has some helpful information, the introduction and method section needs some revision.

1- Please in background, bring some categories of clinical competencies that general nurses should have. You can shorten the content about competency as whole and bring some details of clinical competencies instead. 

2- In method section, please provide more details about sampling framework, and data gathering.

3- In page 4 Table 1 is a demographic table, not the Cronbach's alpha of the questionnaire.

4- Since the attrition is more than 30% there is a possibility of bias in this study, please mention this in your limitations, and if possible please mention whether the subjects that didnt returned the questionnaires were different from those who completed it or not.

5- Please write also the mean and SD of years in service.

6- Please dont repeat the details of the findings in the discussion, just mention for example that competency was more in nurses with 4-10 work experience.

Author Response

Reviewer 3

The manuscript is about the Self-perceived clinical competence of nurses in different clinical settings: A cross-sectional study.

The manuscript has some helpful information, the introduction and method section needs some revision.

1- Please in background, bring some categories of clinical competencies that general nurses should have. You can shorten the content about competency as a whole and bring some details of clinical competencies instead.

  • We thank the reviewer for suggesting; we have inserted a paragraph to specify some general categories of clinical skills.

2- In the method section, please provide more details about the sampling framework and data gathering.

  • We thank the reviewer for this helpful advice, we have included a paragraph in the method section about sampling framework and data gathering.

3- In page 4 Table 1 is a demographic table, not the Cronbach's alpha of the questionnaire.

  • We thank the reviewer for the proposal, we have inserted the relevant table, and we have changed the numbering of the other tables.

4- Since the attrition is more than 30% there is a possibility of bias in this study, please mention this in your limitations, and if possible please mention whether the subjects that didnt returned the questionnaires were different from those who completed it or not.

  • We thank the reviewer for suggesting. We have included a paragraph in the limitations section.

5- Please write also the mean and SD of years in service.

  • We thank the reviewer for the recommendation. We have included the MEAN and SD of years in service.

6- Please dont repeat the details of the findings in the discussion; just mention, for example, that competency was more in nurses with 4-10 years of work experience.

  • We thank the reviewer for the advice. We deleted the paragraph that was redundant compared with the results section.

Reviewer 4 Report

The study is of interest in the field of nursing care. It is well written and structured, with methodology clearly described and appropriate to the type of study. The results are well presented, as well as the discussion and conclusions. However, it requires improvement:

1) in the abstract it is not necessary to number the sections.

2) Line 59: the acronym NCS is not presented. It should mention the full words of its meaning (nursing competency scale) and then the acronym in parentheses, as it is the first appearance in the text.

3) In methods it would help to show the scales in a table, with their scores and categories (lines 131-142).

4) Line 152 mentions Table 1, which does not appear in the text. Then, in results (line 184 mentions Table 1 that appears with the description of the sample studied).

5) The titles of the tables should give an account of what it shows, in whom, when and where. Title of table 1 is too concise.

6) In the conclusions they repeat phrases already said in the section implications for nursing practice (4.2). The conclusions should show the most relevant and interesting results, for example, that the perception of best practice increases with years of experience, among others. Give more detail of the findings with respect to what is mentioned between lines 390 and 396.

Author Response

Reviewer 4

The study is of interest in the field of nursing care. It is well written and structured, with methodology clearly described and appropriate to the type of study. The results are well presented, as well as the discussion and conclusions. However, it requires improvement:

1) in the abstract, it is not necessary to number the sections.

  • We thank the reviewer for the suggestion, we have removed the section numbering in the abstract.

2) Line 59: the acronym NCS is not presented. It should mention the full words of its meaning (nursing competency scale) and then the acronym in parentheses, as it is the first appearance in the text.

  • We thank the reviewer for the suggestion, we have modified line 59 as suggested.

3) In methods it would help to show the scales in a table, with their scores and categories (lines 131-142).

  • We thank the reviewer for the proposal; in table 3, the averages of the scores of the categories were inserted.

4) Line 152 mentions Table 1, which does not appear in the text. Then, in results (line 184 mentions Table 1 that appears with the description of the sample studied).

  • We thank the reviewer for the suggestion, we have inserted the missing table and renumbered the tables within the text.

5) The titles of the tables should give an account of what it shows, in whom, when and where. Title of table 1 is too concise.

  • We thank the reviewer for the advice; we changed the title of the table more descriptively.

6) In the conclusions they repeat phrases already said in the section implications for nursing practice (4.2). The conclusions should show the most relevant and interesting results, for example, that the perception of best practice increases with years of experience, among others. Give more detail of the findings with respect to what is mentioned between lines 390 and 396.

  • We thank the reviewer for the suggestion. We have included two paragraphs in the conclusions, making it more evident to the readers the more crucial implications of the study and what it is mentioned between lines 390-396.

Round 2

Reviewer 1 Report

The authors have revised the manuscript according to the comments. No further comments.

Author Response

Thanks to the reviewer for the suggestions made to improve our manuscript

Reviewer 3 Report

The revisions are acceptable, please merge the tables 4 and 5 since except p value all other data can be seen in table 4.

Author Response

Thanks to the reviewer for his suggestions to improve our manuscript, we have combined Table 4 and Table 5